# Spanish Paediatricians’ Knowledge, Attitudes and Practices Regarding Oral Health of Children under 6 Years of Age: A Cross-Sectional Study

**DOI:** 10.3390/ijerph19159550

**Published:** 2022-08-03

**Authors:** Yolanda Martínez-Beneyto, Celia Navarrete-García, Clara Serna-Muñoz, Veronica Ausina-Márquez, Andrea Poza-Pascual, Antonio Javier Expósito-Delgado, Ascensión Vicente, Antonio J. Ortiz-Ruiz

**Affiliations:** 1Department of Dermatology, Stomatology and Radiology, Faculty of Medicine-Dentistry, Institute of Biomedical Research, IMIB, University of Murcia, 30008 Murcia, Spain; yolandam@um.es; 2Department of Dermatology, Stomatology and Radiology, Faculty of Medicine-Dentistry, University of Murcia, 30008 Murcia, Spain; celia.navarreteg@um.es; 3Department of Integral Paediatric Dentistry, Faculty of Medicine and Dentistry, University of Murcia, 30008 Murcia, Spain; ajortiz@um.es; 4Institute of Biomedical Research, IMIB, 30120 Murcia, Spain; 5Department of Pediatric and Preventive Dentistry, Faculty of Dentistry, European University of Valencia, 46010 Valencia, Spain; veronicaausinamarquez@gmail.com; 6Department of Stomatology I, School of Dentistry, University of the Basque Country, 48940 Lejona, Spain; poza.andrea@gmail.com; 7Andalusian Health Service, 23006 Jaen, Spain; antonioj.exposito.sspa@juntadeandalucia.es; 8Microbiology Section, Department of Biology, Healthcare and Environment, Faculty of Pharmacy and Food Sciences, Universitat de Barcelona (UB), Av Joan XXIII, 27-31, 08028 Barcelona, Spain; ascenvi@um.es

**Keywords:** early childhood caries, oral health, paediatrics, paediatric dentist, oral habits

## Abstract

Background: Early Childhood Caries (ECC) is a prevalent chronic pathology, and it has a negative impact on the oral and general health of the child patient. Aim: To evaluate the knowledge, attitudes and practices of Spanish paediatricians regarding early childhood caries according to the professional’s years of experience. Material and Methods: A cross-sectional questionnaire was conducted by Spanish paediatricians via WhatsApp and e-mails from January to April 2021. Data were analysed using Chi-squared test, Fisher’s exact test and Cramer’s V test. Results: There were a total of 359 participants. Most respondents were women (81.3%) with up to 10 years of professional experience (31.2%) in primary health care and public health. In most cases, participants had an excellent knowledge of primary dentition (90.8%), but they ignored (56%) when the first visit to the dentist should occur. Regarding the aetiological factors of caries, oral hygiene and prevention, a lower rate of knowledge was observed. The majority of participants (80.8%) were not able to identify white spot lesions and enamel defects (76%). They considered that their knowledge in oral health was deficient, highlighting the need to increase their training. Less experienced paediatricians were found to have higher success rates. Conclusions: The level of knowledge and attitudes regarding early childhood caries of the evaluated paediatricians should be improved. Paediatricians had difficulties in identifying early caries lesions and enamel defects. Nevertheless, a higher level of knowledge and positive attitudes towards dental caries has been detected among paediatricians with fewer years of professional experience.

## 1. Introduction

“Early childhood caries is a highly prevalent global disease of public health importance” [1]. This disease represents one of the most aggressive caries patterns among pre-school children. It affects deciduous dentitions and negatively influences the general health and quality of life of the child patient [1,2]. As can be expected, it has a strong influence on the individual and families, and it generates high economic and social costs. However, dental caries is a preventable and reversible disease if treated in its early stages [1].

Dental caries is “a biofilm-mediated, diet-modulated, multifactorial, non-communicable, dynamic disease resulting in net mineral loss of dental hard tissues” [3,4,5]. It is determined by biological, behavioural, psychosocial and environmental factors [5].

Early Childhood Caries (ECC) is defined as “the presence of one or more decayed (non-cavitated or cavitated lesions), missing or filled (due to caries) surfaces, in any primary tooth of a child under six years of age”. In addition, it shares risk factors with other non-communicable diseases such as cardiovascular disease, diabetes and obesity [2]. Early Childhood Caries is considered a major public health problem in both developed and developing countries, with a high prevalence of active caries. According to the latest data published in the Global Burden of Disease Study (2017), it is estimated that more than 530 million children worldwide have caries in the primary dentition. In addition, it is considered the tenth most common chronic disease condition in children [6]. In Spain, the latest Oral Health Survey 2020 shows a prevalence of caries (dmfs/DMFS > 0) in children aged 5–6 years in primary dentition of 35.5%, rising to values above 95% in the United Arab Emirates [7,8].

Early childhood caries has severe consequences including pain, facial infections, hospitalizations, attention deficit, sleep disturbances and psychological impact. This results in a decrease in the child’s quality of life [9,10]. It has been described that children with ECC have a slower growth rate when compared with caries-free children [9].

Dental caries results from dysbiosis in the biofilm environment, caused by the metabolism of fermentable carbohydrates from the diet. The composition of the biofilm can be modified by the conditions of the oral environment, and the changes that occur affect microbial interactions and their relationship with the host [11]. Lack of oral hygiene and inadequate fluoride use are considered primary risk factors in the development of dental caries [2,4].

The role of cariogenic diet is crucial. Early childhood caries is closely related to frequent consumption of fermentable carbohydrates, mainly sucrose [2]. The World Health Organization (WHO) recommends that refined sugars should not be offered to children before the age of 2 years, and the amount of free sugars should be no more than 5% of their energy intake. A child’s diet should be rich in fresh produce, healthy and balanced [1].

Paediatricians are the first professionals to assess the child patient. They follow the patient’s growth and development closely [12,13]. Furthermore, the family often have a relationship of trust with the professional. Literature suggests that children under 3 years of age have visited a paediatrician an average of eleven times since birth [14]. However, most studies agree that few children aged 0–3 years visit the dentist, and it is estimated that the first visit usually occurs from the age of 4.5 years onwards [14,15].

Therefore, collaboration with paediatricians is essential, as they can detect oral diseases early and refer the patient to a paediatric dentist [13].

The aim of this study was to evaluate the knowledge, attitudes and practices regarding oral health, hygiene habits and early childhood caries of Spanish paediatricians according to years of experience. In addition, the ability to identify carious lesions in young children was assessed.

## 2. Material and Methods

This article was written in accordance with the STROBE Statement for cross-sectional studies [16]. The study was approved by the Bioethics Committee of Murcia University (Reference Number: 3234/2021).

### 2.1. Sample and Procedure

The study population consisted of Spanish paediatrics belonging to five Spanish paediatric associations: Sociedad de Pediatría del Sureste (Murcia), Sociedad de Pediatría de Murcia, Sociedad Vasco Navarra de Pediatría, Sociedad Valenciana de Pediatría and paediatricians from the health areas of Jaén.

All the members of the associations were invited to participate in the study. They received a link, by WhatsApp and by email, to the University of Murcia’s Survey Platform (https://encuestas.um.es/encuestas/saludoralpediatras.cc, accessed on 19 February 2021), in order to complete the questionnaire. The link was active from 10 January until 25 April 2021. All participants were informed in the message that the survey was anonymous and that its completion implied consent to participate in the study. Two separate sampling rounds were carried out in two months to increase participation.

A pilot study was carried out with the participation of 25 paediatricians from different regions to ensure that it was correctly designed, the questions were well understood and it did not require a long time to answer.

### 2.2. Study Instrument

A questionnaire was designed using the University of Murcia’s Survey Platform (https://encuestas.um.es/, accessed on 19 February 2021). It was designed based on the studies of Caspary et al. [17] Subramaniam et al. [18], Balaban et al. [19], Kalkani et al. [20] Ramroop et al. [14] Alshunaiber et al. [21] and Gupta et al. [13].

The questionnaire was divided into five main sections:Section I. Socio-demographic characteristics.

Seven questions relating to gender, age, professional experience, region and work sector.

Section II. Knowledge and attitudes regarding primary dentition.

Seven questions on eruption, tooth brushing, first visit to the paediatric dentist and oral health education measures.

Section III. Paediatrician’s knowledge and attitudes towards oral health and early childhood caries.

Twelve questions related to the current concept of dental caries, the aetiological factors of the disease and preventive measures.

Section IV. Enamel lesion identification.

Six questions, four of which were illustrated with clinical photographs of caries or enamel defects; the paediatrician had to identify the correct lesion.

Section V. Training.

Four questions for practitioners to self-assess their training in relation to oral health.

### 2.3. Data Analysis

Study data were processed and analysed using the R Core Team statistical package (free software). A simple frequency distribution was made for variables. To identify significant differences and search for relationships based on sociodemographic variables, Fisher’s exact test was applied. Cramer’s test was used to determine whether the relationship was strong or weak.

## 3. Results

### 3.1. Descriptive Analysis

A total of 2580 paediatricians were invited to participate in the study, but only 359 responded (response rate: 13.9%). The majority of the participants were female (81.3%), aged 30–39 years, with up to 10 years of professional experience (31.2%), and they worked mainly in urban areas (75.2%) (Table 1).

Most of the participants (90.8%) knew the eruption time of the primary dentition as well as the total number of teeth in the mouth (83.8%). Regarding the first preventive measures, 57.4% of the paediatricians indicated that they should be implemented after the eruption of the first tooth, and the first visit to the paediatric dentist should take place in the first year of the child’s life (44%). Finally, 83% of respondents stated that they routinely examined their baby’s teeth from the first year of life onwards. (Table 2).

Overall, 48.5% of the paediatricians think that dental caries is a transmissible disease, and that the aetiological factors are diet and bacteria (98.6%), heredity (78.6%), fluoride (73.3%), time of exposure to sugars (97.2%) and saliva (72.7%). In addition, 63.5% of the paediatricians give information to parents about the cariogenic effects of going to bed with a bottle. The majority (67.7%) of respondents agreed that prolonged breastfeeding is not an aetiological factor in the development of early childhood caries (Table 3).

In terms of oral hygiene, 35.7% of the paediatricians always and routinely instruct their patients on how to brush their teeth, and 60.4% acknowledged that they recommend parents to supervise brushing until 7–8 years of age. Furthermore, 64.3% always recommend the use of fluoride toothpaste, but only 52.9% with the correct concentration of 1000 ppm F (Table 4).

Overall, 85.2% of the respondents stated that they always routinely check their patients’ mouths. The majority (96.1%) of the paediatricians identified dental plaque as an accumulation of whitish or yellow plaque.

With regard to the identification of lesions of carious and non-carious origin, Table 5 shows the distribution of responses obtained. The answers in bold indicate the correct diagnostic options.

Considering the level of training received in oral health, 79.4% of the respondents considered it to be deficient. Only 9.2% of paediatricians stated that they had received training in paediatric dentistry during their residency.

Almost all respondents (98.1–98.3%) acknowledged the need for oral health training during the medical degree and the speciality of paediatrics.

### 3.2. Inferencial Analysis

Table 6 shows the association between the level of knowledge, attitudes and practices of the professionals surveyed and the years of professional experience.

In general, both the level of knowledge, attitudes and practices of professionals with 10–20 years of professional experience were associated (*p* < 0.05) with a higher hit rate (Table 6).

## 4. Discussion

Despite the high prevalence of early childhood caries in Spain, this is the first study to be carried out in our country.

A low response rate (13.9%) was obtained even though the survey was sent twice, separated by a reasonable time interval, and using similar methods to other studies [22] The response rate has long been observed as a measure of quality of work, but no minimum response rate has been scientifically accepted.

Non-response bias, meaning that respondents do not represent the target population of the study, is more of a problem in general population surveys than in specific groups such as paediatricians [23]. The questionnaire was to be completed via an online link because of the ease of reaching the target population, as well as the COVID-19 pandemic situation. Other studies combine the online system with the distribution of printed copies [14,20,21]. The sampling model used in this study was expert sampling, a non-probabilistic purposive sampling method [24].

In Spain, practically all paediatricians work in public centres of the National Health System, with the vast majority in primary health care centres. This has been represented in the study. It is in health centres where health education is carried out and where there is the opportunity to work in a multicentric way with paediatricians and paediatric nurses in terms of oral health.

In terms of the results obtained, most paediatricians (90.8%) knew the time of eruption of the first primary teeth; however, they did not know when the first visit to the dentist should take place, which coincides with the study carried out in Saudi Arabia [21] According to the recommendations of the American Association of Paediatric Dentistry, the first dental visit should occur during the first year of life or when the first primary tooth erupts. In our study, only 44% of the participants recommended the first dental visit at 1 year of age, and 46.2% postponed this visit until 3 years of age. Other studies have obtained similar results, such as one conducted in India [25] or another in the United States in 2008, where 50% of the participants responded that the first dental visit should occur at 3 years of age [14].

The majority of paediatricians (83%) examined the infant’s teeth from the first year of life, and most provided oral health education to parents to prevent dental caries, as consistent with data from studies by Lewis et al. [26], Balaban et al. [19] and Gupta et al. [13]. Paediatricians often only refer children to a paediatric dentist at the request of the parents, in agreement with other studies [13]. However, they did not refer for preventive reasons, and most organisations agree that preventive measures should be applied from the eruption of the first primary tooth, coinciding with the first visit to the dentist [2]. There are still many specialists who understand caries as a transmissible disease, centred on the concept of bacterial infection. However, it is now known that caries is a non-communicable disease, and the final objective is to seek an oral microbiological balance [19].

The majority of respondents correctly recognised most of the aetiological factors, especially diet, bacteria and time of exposure to sugars. The studies by Balaban et al. [19], Kalkani et al. [20] and Ramroop et al. [14] obtained similar results. The least cited factors in the study were protective factors such as fluoride and saliva. Night-time consumption of sugars has been shown to increase the risk of developing early childhood caries because oral hygiene measures are lacking, salivary defence mechanisms are at rest and bacteria rapidly increase acid production, demineralising tooth enamel [3]. In this sense, many Spanish paediatricians recommend not using a bottle at night. Breastfeeding and its relationship to dental caries has recently been discussed by professional organisations. Studies have linked dental caries in infants older than 12 months to prolonged breastfeeding [27]. The increased caries risk in these cases is probably due to the combination of breastfeeding with sugary complementary foods and the absence of oral hygiene. In our study, 67.7% of paediatricians stated that prolonged breastfeeding was not an aetiological factor, showing very similar values to those described in the study conducted by Gupta et al. [13]. Breastfeeding is currently considered a protective factor against caries. It also promotes the growth of the patient by exercising the masticatory muscles and strengthens the immune system of the paediatric patient through the transmission of maternal immunological components, and should be maintained up to two years of age [27].

In our study, only 35.7% routinely indicated how to brush their patients’ teeth and supervised brushing by parents up to 7–8 years of age. The study by Prathima et al. [12] obtained similar results, and in addition, only 39.1% of their respondents recommend supervised tooth brushing from the eruption of the first tooth. In our study, this value is slightly higher (57.4%). This situation confirms the lack of oral health education of health professionals not only in Spain, but also in other countries of the world.

The role of fluoride as a primary preventive measure against dental caries is widely supported in the literature, and the tendency of paediatricians to recommend lower doses or restrict its use may be due to the risk of fluorosis or toxicity [2,4,28]. However, a fluoride concentration in toothpastes for children under 6 years of age of between 1000–1450 ppm, as well as the application of fluoride varnish periodically by the practitioner, has been shown to outweigh the benefits against the risks exposed [28]. Toothpaste should be fluoridated, with a minimum of 1000 ppm F, avoiding subsequent rinsing with water [29,30]. This information was known to only 52.9% of the paediatricians surveyed. In their study, Ramroop et al. [14] reported that 22.4% of paediatricians did not advise parents to use fluoride toothpastes for children aged 0–3 years, and 49.9% recommended a concentration of 250 ppm fluoride. In Spain, there are still professionals who have stated that they do not recommend fluoride toothpaste for children up to certain ages.

With regard to the identification of caries lesions, in our study, the majority of professionals (80.8%) did not identify the white spot lesion in the cervical third of a primary tooth. This situation also occurs in countries such as Brazil [19], the United Kingdom [20] and Trinidad and Tobago [14] where only a small percentage of paediatricians can identify such a lesion.

The prevalence of enamel defects in primary dentition and their relationship with dental caries has been demonstrated in the literature [31]. This situation would justify an early diagnosis by the paediatrician for subsequent referral to a dentist. However, in our study, only 24% of paediatricians were able to diagnose enamel defect lesions in molars, compared to 79.1% in anterior teeth.

Paediatricians stated that they need to increase their knowledge of oral health. Most of the professionals considered their oral health training to be deficient and highlighted the need to expand their knowledge both during their medical degree and throughout the residency period of their speciality. The study by Herndon et al. (2015) identified an increase in confidence in professionals who receive oral health training, which positively influences the transmission of knowledge to parents about early childhood caries prevention [32].

Finally, it should be noted that paediatricians with less professional experience were more trained in oral health and early childhood caries than those with more extensive experience, as reflected in the results of our study.

The main limitation of the study could be the low response rate of paediatricians, despite two rounds of contact.

## 5. Conclusions

To conclude, the level of knowledge and attitudes regarding early childhood caries of the evaluated paediatricians should be improved in order to prevent caries disease and to improve the oral health of the child patient.

In this study, paediatricians had difficulties in identifying early caries lesions and enamel defects.

Finally, a higher level of knowledge and positive attitudes towards dental caries has been detected among paediatricians with fewer years of professional experience; however, there is a need for further training of paediatricians in oral health.

The incorporation of oral health contents in paediatric training curricula would be necessary to increase knowledge and positive attitudes towards the most frequent pathology among children.

## Figures and Tables

**Table 1 ijerph-19-09550-t001:** Demographic characteristics of participants.

Demographics	Total n (%)
**Gender**	
Male	67 (18.7)
Female	292 (81.3)
**Age group**	
21–29 years	24 (6.7)
30–39 years	130 (36.2)
40–49 years	68 (18.9)
50–59 year	86 (24.0)
>60 years	51 (14.2)
**Years in practice**	
<10 years	112 (31.2)
11–20 years	101 (28.1)
21–30 years	85 (23.7)
31–40 years	52 (14.5)
>40 years	9 (2.5)
**Regions**	
Andalucía	77 (21.4)
Comunidad Valenciana	106 (29.5)
País Vasco	64 (17.8)
Region of Murcia	112 (31.2)
**Area**	
Rural	30 (8.4)
Semi-urban	59 (16.4)
Urban	270 (75.2)
**Work Sector**	
Public	342 (95.3)
Private	17 (4.7)
**Type of practice**	
Primary care	268 (74.6)
Hospital	91 (25.3)
**Total**	359 (100)

**Table 2 ijerph-19-09550-t002:** Knowledge and attitudes about primary dentition.

Questions	Participants n (%)
**When does the first primary tooth erupt?**	
4–5 months	29 (8.1)
6–8 months	326 (90.8)
12 months	3 (0.8)
15 months	1 (0.3)
**Number of primary teeth:**	
25	6 (1.7)
28	18 (5.1)
20	301 (83.9)
18	34 (9.5)
**When should you brush your child’s teeth with fluoride toothpaste?**	
As soon as the first tooth erupts	206 (57.4)
When several teeth have erupted	27 (7.5)
When the child is able to use the brush	99 (27.6)
I am not sure	27 (7.5)
**First visit to the dentist:**	
6 months of age	22 (6.1)
1 years old	158 (440)
When the first cavities are pesent	13 (3.6)
3 years old	166 (46.2)
**Do you examine the patient’s teeth from the first year of life?**	
Always and routinely	298 (83.0)
Only if pain is reported	5 (1.4)
Occasionally	51 (14.2)
Never	5 (1.4)
**Do you provide oral health education to parents?**	
Always and routinely	274 (76.3)
Occasionally	74 (20.6)
Never	11 (3.1)
**When do you refer your patients to a paediatric dentist?**	
Always and routinely	174 (48.5)
When parents indicate dental problems	181 (50.4)
Never	4 (1.1)

**Table 3 ijerph-19-09550-t003:** Dental caries and aetiological factors.

Questions	Participants n (%)	
**Is dental caries a transmissible disease?**		
Yes	174 (48.5)	
No	159 (44.3)	
I do not know	26 (7.2)	
**Aetiological factors**: (multiple choice)	Yes	No
Diet and bacteria	354 (98.6)	5 (1.4)
Hereditary factor	282 (78.5)	77 (21.5)
Fluoride	263 (73.3)	96 (26.7)
Sugar exposure time	349 (97.2)	10 (2.79)
Saliva	261 (72.7)	98 (27.3)
**Restrictions on sugary foods**: (multiple choice)	Yes	No
No snacking	216 (60.2)	143 (39.8)
Do not drink non-natural juices	290 (80.8)	69 (19.2)
Avoid sugary drinks	335 (93.3)	24 (6.7)
Avoid sweets	326 (90.8)	33 (9.2)
No	1 (0.3)	358 (99.7)
**Do you remark on the cariogenic effects of going to bed with a bottle?**		
Always and routinely	228 (63.5)	
Occasionally	115 (32.0)	
Never	16 (4.5)	
**Do you believe that breastfeeding is an aetiological factor in tooth decay?**		
Yes	81 (22.6)	
No	243 (67.7)	
I do not know	35 (9.7)	

**Table 4 ijerph-19-09550-t004:** Oral hygiene and preventive measures.

Questions	Participants n (%)
**Do you indicate how to brush your patients’ teeth?**	
Always and routinely	128 (35.6)
Occasionally	171 (47.6)
Never	60 (16.7)
**Up to what age should parents supervise brushing?**	
Until the child can do it alone	123 (34.3)
Up to 3 years of age	13 (3.6)
Up to 7–8 years of age	217 (60.4)
Never	6 (1.7)
**Do you recommend the use of dental floss?**	
Always and routinely	30 (8.4)
Occasionally	102 (28.4)
Never	117 (32.6)
Depends on age	110 (30.6)
**Do you recommend fluoride toothpaste?**	
Always and routinely	231 (64.3)
Occasionally	29 (8.1)
Never	8 (2.2)
Depends on age	91 (25.3)
**For children under 6 years of age, do you recommend the use of fluoride mouth rinses?**	
Always and routinely	10 (2.8)
Occasionally	73 (20.3)
Never	241 (67.1)
Depends on age	35 (9.7)
**For children under 6 years of age, what concentration of fluoride toothpaste do you recommend?**	
1450 ppm F	36 (10.0)
1000 ppm F	190 (52.9)
500 ppm F	41 (11.4)
I do not know	92 (25.6)
**Do you prescribe fluoride tablets or drops for children under 3 years of age?**	
Always	0 (0.0)
Occasionally	21 (5.8)
Never	338 (94.1)

**Table 5 ijerph-19-09550-t005:** Identification of lesions by clinical imaging.

		N (%)
**IMAGE 1** 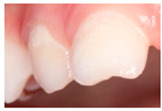	**Caries**	**69 (19.2)**
	Enamel defect	249 (69.4)
	No dental lesions	15 (4.2)
	I do not know	26 (7.2)
**IMAGE 2** 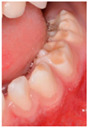	Dental caries	244 (68.0)
	**Enamel defects**	**86 (24.0)**
	No dental lesions	8 (2.2)
	I do not know	21 (5.8)
**IMAGE 3** 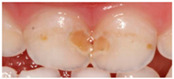	**Dental caries**	**294 (81.9)**
	Enamel defects	53 (14.8)
	No dental lesions	12 (3.3)
	I do not know	0 (0.0)
**IMAGE 4** 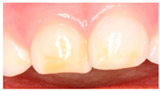	Dental caries	40 (11.1)
	**Enamel defects**	**284 (79.1)**
	No dental lesions	13 (3.6)
	I do not know	22 (6.1)

**Table 6 ijerph-19-09550-t006:** Knowledge of primary dentition, oral health and early childhood caries and lesion identification according to years of experience of the study sample.

	Years of Professional Experience n (%)					
Questions	≤10 Years	11–20 Years	21–30 Years	31–40 Years	≥40 Years	*p* Value *
**Brushing with fluoride toothpaste:**						
First tooth eruption	86 (24.0)	66 (18.4)	34 (9.5)	19 (5.3)	1 (0.3)	*p* < 0.05
**First visit to the dentist:**						
1 year	8 (2.2)	10 (2.8)	4 (1.1)	0	0	*p* < 0.05
3 years	64 (17.8)	40 (11.1)	33 (9.2)	18 (5.0)	3 (0.8)	
**Do you examine the patient’s teeth from the first year of life?**						
Always and routinely	82 (22.8)	81 (22.6)	77 (21.4)	50 (14.0)	8 (2.2)	*p* < 0.05
**Do you provide oral health education to parents?**						
Always and routinely	72 (20.1)	75 (20.9)	75 (20.9)	46 (12.0)	6 (1.7)	*p* < 0.05
**When do you refer your patients to a paediatric dentist?**						
Always and routinely	38 (10.6)	54 (15.0)	49 (13.6)	30 (8.4)	3 (0.8)	*p* < 0.05
Dental problems	73 (20.3)	47 (13.1)	35 (9.7)	21 (5.8)	5 (1.4)	
**Is dental caries a communicable disease?**						
Yes	55 (15.3)	60 (16.7)	33 (9.2)	25 (7.0)	1 (0.3)	*p* < 0.05
No	47 (13.1)	33 (9.2)	46 (12.8)	27 (7.5)	6 (1.7)	
**Aetiological factors:** (multiple choice)						
Diet and bacteria	112 (31.2)	100 (27.9)	84 (23.4)	50 (13.9)	8 (2.2)	*p* < 0.05
Fluoride	92 (25.6)	80 (22.3)	51 (14.2)	35 (9.7)	5 (1.4)	
Saliva	86 (24.0)	84 (23.4)	60 (16.7)	29 (8.1)	2 (0.6)	
**Do you remark on the cariogenic effects of going to bed with a bottle?**						
Always and routinely	50 (13.93)	64 (17.8)	66 (18.4)	42 (11.7)	6 (1.7)	*p* < 0.05
**Do you recommend fluoride toothpaste?**						
Always and routinely	84 (23.4)	70 (19.5)	48 (13.4)	26 (7.2)	3 (0.8)	*p* < 0.05
**For children under 6, do you recommend the use of fluoride mouth rinses?**						
Never	79 (22.0)	71 (19.8)	52 (14.5)	34 (9.5)	5 (1.4)	*p* < 0.05
**For children under 6, what concentration of fluoride toothpaste do you recommend?**						
1000 ppm F	70 (19.5)	58 (16.2)	39 (10.9)	21 (5.8)	2 (0.6)	*p* < 0.05
**Do you prescribe fluoride tablets or drops for children under 3 years of age?**						
Never	107 (29.8)	99 (27.6)	77 (21.4)	49 (13.6)	6 (1.7)	*p* < 0.05
**IMAGE 3**						
Caries	102 (28.4)	83 (23.1)	66 (18.4)	37 (10.3)	6 (1.7)	*p* < 0.05
**IMAGE 4**						
Enamel defects	92 (25.6)	83 (23.1)	60 (16.7)	43 (12.0)	6 (1.7)	*p* < 0.05

* Fisher’s exact test.

## Data Availability

The datasets used for the current study are available from the corresponding author upon reasonable request.

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
