# Peer review of "Spanish Paediatricians’ Knowledge, Attitudes and Practices Regarding Oral Health of Children under 6 Years of Age: A Cross-Sectional Study"

_ijerph, 2022, doi:10.3390/ijerph19159550_

Round 1

Reviewer 1 Report

Introduction: In Spanish,  before, are there many research regarding the knowledge, attitudes and practices on oral health, hygiene habits in Spanish paediatricians under 6 years? Can you add somes result regarding these researchs?

In table 1: the pevalence of Work Sector in public (95.3%) is very different with the Work Sector in private (4.7%), also, it’s difficult to compare the difference between the public and private.

Similary, regarding the type of practice, primary care is 74.6%, hospital is 25.3%. Also , you can need to discus this issue can effect or no effect the result?

Author Response

Dear reviewer, we have answered the minor questions you have suggested.

  1. Introduction: In Spanish,  before, are there many research regarding the knowledge, attitudes and practices on oral health, hygiene habits in Spanish paediatricians under 6 years? Can you add somes result regarding these researchs?

Response: Dear Reviewer, In Spain there is no other research that evaluates knowledge, attitudes and practices on oral health, hygiene habits in Spanish paediatricians under 6 years

  1. In table 1: the pevalence of Work Sector in public (95.3%) is very different with the Work Sector in private (4.7%), also, it’s difficult to compare the difference between the public and private.

Response: In Spain, fortunately, paediatric care is mostly provided in hospitals and primary care centres (National Health Services). Although training in the speciality is public for all paediatricians in Spain. So this information was just to know the prevalence of paediatricians working in private clinics.

  1. Similary, regarding the type of practice, primary care is 74.6%, hospital is 25.3%. Also , you can need to discus this issue can effect or no effect the result?

Response: As mentioned above, the majority of care is provided in the public sector and in primary health care, a situation that is reflected in the descriptive data presented below. A paragraph has been added in the discussion which explains this situation.

“In Spain, practically all paediatricians work in public centres of the National Health System, and the vast majority in primary health care centres. These results have been represented in the study. It is in health centres where health education is carried out and where there is the opportunity to work in a multicentric way with paediatricians and paediatric nurses in terms of oral health”

Reviewer 2 Report

Congratulations on your manuscript, it is very interesting.

There are a couple of remarks to improve its quality:

-       This situation could identify a higher level of knowledge among newly graduated dentists, where there is an  updated European and Spanish cariology curriculum.

The study is focus in paediatricians. Please, remove or change this sentence. 

-       Include a figure that summarizes the results part as much as possible. 

- Please, review all the text editing (there are many typographic errors)

Author Response

REVIEWER 2.
1. The study is focus in paediatricians. Please, remove or change this sentence.
Response: Lines 260-2 have been removed.
2. Include a figure that summarizes the results part as much as possible.
Response: Dear reviewer, the results are summarized in 4 tables, following the sections of the survey, table 5 with the images and finally table 6 with the association between variables. It is a spread survey and the data have been summarized as much as possible.
3. Please, review all the text editing (there are many typographic errors)
Response: We have reviewed the manuscript.

Reviewer 3 Report

dear authors, I really appreciate the main theme of the paper. Some adjustment are needed:

- some typos are presente in both abstract and main text, please correct

Introduction

- I suggest removing the inverted commas and rephrasing the sentence to make it more discursive with an appropriate bibliographic citation

- "Global Burden of Disease Study (2017)" Published data are now outdated, 2019 data are available

- lines 46-48 I suggest comparing the Spanish data with European countries with a greater affinity to the Spanish population, e.g. prevalence data in Italy

- line 50-51. there are several questionnaires aimed at assessing the impact of oral health on general health (e.g. the ECOHIS questionnaire). i suggest a little background on this

materials and methods

- please define how the appropriate sample size was calculated or whether a convenience sample was used instead

Results

-the tables are difficult to decipher, I suggest not setting the text centred and not heading, each item must have its result in the same line

-what is written in the tables should not be repeated in the text

-results should always be expressed with two decimals after the point and not the comma, please standardise all numerical data

- I only see the results of the fisher exact, when was chi square used? no results derived from the Cramer test or chi square are given, please clarify

discussion

- lines 168-169 the sentence is very strong and misunderstood, please rephrase

- line 196-199, the sentence is not clear, please rephrase

- line 200, is a correct thought but detached from the text, please elaborate on how the aetiological paradigm of caries has changed and how it has become a non-communicable disease in recent years

- "in our study" please rephrase impersonal

- line 239, it is very difficult to determine lesion activity from a photo, please define it correctly as a white spot lesion

- in discussion I suggest solutions to the results found to be further investigated in future studies, how can the theoretical knowledge and diagnostic-clinical skills of paediatricians be improved? on the basis of the results which fields should be improved the most?

Author Response

The attached document are the responses to reviewer 1. The other modification are made in main manuscript. Sorry for the inconvenience and thank you
